# Chemically Activated Cooling Vest’s Effect on Cooling Rate Following Exercise-Induced Hyperthermia: A Randomized Counter-Balanced Crossover Study

**DOI:** 10.3390/medicina56100539

**Published:** 2020-10-14

**Authors:** Yuri Hosokawa, Luke N. Belval, William M. Adams, Lesley W. Vandermark, Douglas J. Casa

**Affiliations:** 1Faculty of Sport Sciences, Waseda University, Saitama 359-1192, Japan; 2Institute for Exercise and Environmental Medicine, Texas Health Presbyterian Hospital Dallas and University of Texas Southwestern Medical Center, Dallas, TX 75231, USA; lukebelval@texashealth.org; 3Department of Kinesiology, University of North Carolina at Greensboro, Greensboro, NC 27412, USA; wmadams@uncg.edu; 4Department of Health, Human Performance, and Recreation, University of Arkansas, Fayetteville, AR 72701, USA; lwvander@uark.edu; 5Korey Stringer Institute, Department of Kinesiology, University of Connecticut, Storrs, CT 06269, USA; douglas.casa@uconn.edu

**Keywords:** exertional heat stroke, emergency treatment, ice vest, body cooling, prehospital care

## Abstract

*Background and objectives:* Exertional heat stroke (EHS) is a potentially lethal, hyperthermic condition that warrants immediate cooling to optimize the patient outcome. The study aimed to examine if a portable cooling vest meets the established cooling criteria (0.15 °C·min^−1^ or greater) for EHS treatment. It was hypothesized that a cooling vest will not meet the established cooling criteria for EHS treatment. *Materials and Methods:* Fourteen recreationally active participants (mean ± SD; male, *n* = 8; age, 25 ± 4 years; body mass, 86.7 ± 10.5 kg; body fat, 16.5 ± 5.2%; body surface area, 2.06 ± 0.15 m^2^. female, *n* = 6; 22 ± 2 years; 61.3 ± 6.7 kg; 22.8 ± 4.4%; 1.66 ± 0.11 m^2^) exercised on a motorized treadmill in a hot climatic chamber (ambient temperature 39.8 ± 1.9 °C, relative humidity 37.4 ± 6.9%) until they reached rectal temperature (T_RE_) >39 °C (mean T_RE_, 39.59 ± 0.38 °C). Following exercise, participants were cooled using either a cooling vest (VEST) or passive rest (PASS) in the climatic chamber until T_RE_ reached 38.25 °C. Trials were assigned using randomized, counter-balanced crossover design. *Results:* There was a main effect of cooling modality type on cooling rates (F[1, 24] = 10.46, *p* < 0.01, η^2^_p_ = 0.30), with a greater cooling rate observed in VEST (0.06 ± 0.02 °C·min^−1^) than PASS (0.04 ± 0.01 °C·min^−1^) (MD = 0.02, 95% CI = [0.01, 0.03]). There were also main effects of sex (F[1, 24] = 5.97, *p* = 0.02, η^2^_p_ = 0.20) and cooling modality type (F[1, 24] = 4.38, *p* = 0.047, η^2^_p_ = 0.15) on cooling duration, with a faster cooling time in female (26.9 min) than male participants (42.2 min) (MD = 15.3 min, 95% CI = [2.4, 28.2]) and faster cooling duration in VEST than PASS (MD = 13.1 min, 95% CI = [0.2, 26.0]). An increased body mass was associated with a decreased cooling rate in PASS (r = −0.580, *p* = 0.03); however, this association was not significant in vest (r = −0.252, *p* = 0.39). *Conclusions:* Although VEST exhibited a greater cooling capacity than PASS, VEST was far below an acceptable cooling rate for EHS treatment. VEST should not replace immediate whole-body cold-water immersion when EHS is suspected.

## 1. Introduction

Exertional heat stroke (EHS) is a potentially lethal, hyperthermic condition that is characterized by internal body temperature exceeding 40.5 °C with concurrent central nervous dysfunction [1]. EHS patients must be cooled rapidly using a treatment modality that can achieve an optimal cooling rate of ≥0.15 °C·min^−1^ to ensure patient survival and prevent organ damage from prolonged (≥30 min) heat stress [2]. The current standard of care for cooling EHS patients is whole-body cold-water immersion, which is reported to have a cooling rate of ≈0.20 °C·min^−1^ [2,3,4]. Evidence supporting the use of whole-body cold- water immersion is the 100% survival rate of EHS among 274 runners (age, 13–65 years; initial rectal temperature [T_RE_], 40.0–42.8 °C) succumbing to this medical emergency during a summertime road race. [5] The superiority of whole-body cold-water immersion as the method of EHS treatment is supported by its ability to maximize convective and conductive body heat loss over a large body surface area at once [3,6,7].

When access to ample amounts of water and ice are limited, one may consider cooling EHS patients using a different method [7]. For example, a case series by Gomm et al. [8] used a combination of a cooling vest (CAERvest^®^, BodyChillz Ltd., East Finchley, UK) and ice packs or cooled intravenous infusion to treat three EHS runners (Table 1). The reported average cooling rate in this case series was 0.13 °C·min^−1^, with one patient (Table 1, Patient 3) experiencing rhabdomyolysis, intestinal ischemia, and bilateral compartment syndrome due to prolonged hyperthermia. [8] While this average cooling rate reported by Gomm et al. [8] is greater than cooling rates reported in previous studies that investigated other models of cooling vests (≤0.05 °C·min^−1^) [3,7,9], it does not meet the optimal threshold (≥0.15 °C·min^−1^) for effective body cooling in EHS treatment. These findings suggest that clinicians should avoid the use of a cooling vest as the primary method of EHS treatment, despite its convenience in preparation and application.

Nevertheless, previous cross-over studies that investigated the effectiveness of cooling vests were conducted in individuals with mild hyperthermia (T_RE_ < 39 °C) [7,9] or normothermic individuals (T_RE_
≈ 37 °C) [10]. The effectiveness of cooling vests in cooling individuals with exercise-induced hyperthermia (>39.0 °C) is limited solely to the case series reported by Gomm et al. [8]. However, the cooling rates reported in the aforementioned case series utilized a combination of CAERvest^®^ and ice packs or intravenous infusion, which does not allow for the independent assessment of the cooling capacity offered by the CAERvest^®^ in hyperthermic (>39.0 °C) individuals. Moreover, the application of CAERvest^®^ in previous reports was limited to male patients [8] and subjects [10]. Therefore, the generalizability of the device’s cooling rate on female and individuals with varying anthropometric characteristics needs to be evaluated. It is well acknowledged that body-surface-area-to-body-mass ratio can influence the efficiency of body heat exchange [11]. Individuals with lower body-surface-to-mass ratio (i.e., large persons) may not benefit as much from a standardized sized cooling vest due to greater heat storage [11]. Furthermore, since cooling vests rely on convection as the means of heat transfer, the relative amount of body surface area covered by the vest can directly influence the cooling outcome.

The primary aim of this study was to examine the cooling rate of a cooling vest (CAERvest^®^) in comparison to passive rest on individuals with exercise-induced hyperthermia (>39.0 °C). The secondary aim of this study was to determine if anthropometric characteristics differences observed between sex (body mass, body fat, body surface area) influenced the cooling rate using a cooling vest. It was hypothesized that the cooling rate of the cooling vest would exceed the cooling rate of passive rest. Further, it was hypothesized that the cooling vest would not meet the established cooling criteria (0.15 °C·min^−1^ or greater) [3] for EHS treatment.

## 2. Materials and Methods

### 2.1. Study Design and Setting

This laboratory study utilized a randomized, counter-balanced crossover design of two exercise sessions, each followed by a randomly assigned cooling condition. All trials, including post-exercise cooling, were conducted in a climate-controlled chamber (Model 2000, Minus-Eleven, Inc., Malden, MA, USA); ambient temperature, 39.8 ± 1.9 °C; relative humidity, 37.4 ± 6.9%).

### 2.2. Selection of Participants

Eligibility criteria for the study consisted of individuals who were recreationally active, who engage in at least 30 min of exercise 4–5 days per week. All female participants participated during the luteal phase of their menstrual cycle to account for differences in basal body temperature throughout the menstrual cycle, which was identified from their self-reported menstrual history [12]. Participants were excluded from the study if they reported a history of cardiovascular, metabolic, or respiratory disease, and history of EHS. Participants were free from fever and illness, or a musculoskeletal injury that limited physical activity. All participants provided written informed consent to participate in this study, where ethics approval was granted by the Institutional Review Board at the University of Connecticut (Protocol ID H15-153). Recruitment and data collection occurred between the dates of 18 June 2015 and 21 April 2016.

### 2.3. Interventions

During a familiarization session, participants were instructed on the exercise and cooling procedures utilized during each testing session in the study, provided a baseline body mass measurement on a scale to the nearest 0.01 kg (Defender 5000, OHAUS, Parsippany, NJ USA), and a trained researcher obtained body fat measurements using a three-site skin caliper method (Lange Skinfold Caliper, Cambridge, MD, USA) [13,14]. The two experimental testing sessions were scheduled at least 24 h apart from each other. YH used a computerized random number generator to assign each participant to a cooling condition: (1) cooling vest (CAERvest^®^, BodyChillz Ltd., East Finchley, UK) (VEST) or (2) passive cooling (PASS).

Prior to each testing session, all participants were instructed to consume an extra 500 mL of water the night before and in the morning of their scheduled visit to the laboratory to encourage a state of euhydration at the start of exercise, which was operationally defined as urine-specific gravity ≤1.020 [15]. Urine-specific gravity was measured using light refractometer (Atago Inc., model A300CL, Spartan, Tokyo, Japan). When the urine-specific gravity was >1.020, the participant was asked to consume 500 mL of water to ensure proper hydration status. Water intake was restricted during exercise sessions.

Body temperature was taken via a rectal thermometer (Model 401, Measurement Specialties, Hampton, VA, USA) inserted 10 cm past the participant’s anal sphincter. Heart rate was taken via a heart rate monitor (Race Trainer, Timex Group USA, Middlebury, CT, USA) worn around the participant’s chest. Prior to entering the climate-controlled chamber for both exercise sessions, participants donned a short-sleeved T-shirt, sports bra (female only), running shorts, socks, and sneakers. Upon entering the chamber, participants took a seated position for ten minutes in order to acclimate to the environmental conditions. The exercise portion of each testing session consisted of a 20-min exercise sequence on a motorized treadmill; a 5-min walk between 5.6 and 7.2 km·h^−1^ at a 5% incline, followed by a 15-min jog between 8.9 and 12.1 km·h^−1^ at a 1% incline, repeated up to 3 times. Participants were instructed to freely select their exercise intensity from the predetermined range to ensure that the exercise intensity was sufficient to induced exercise-induced hyperthermia while also considering the participant comfort. All exercise protocol was conducted under direct supervision of at least one medical personnel. Participants were allowed to stop at any point during the 60-min exercise session once their T_RE_ exceeded 39 °C.

Upon cessation of exercise, participants removed their T-shirt and shoes, and the cooling portion of the testing session began. The cooling intervention for both conditions took place within the climate-controlled chamber to replicate the scenario of pre-hospital care (i.e., providing the care onsite in the heat). In the VEST condition, participants laid supine on a cot, while CAERvest^®^ was activated using an external water reservoir kept at room temperature (3000 mL), which triggers the chemicals inside the vest resulting in a cooled vest surface (Figure 1C). The dimension of vest was 730 mm by 820 mm, and covered the ventral side of torso from shoulder to hip (Figure 1D). After each trial, the vest was discarded. During the PASS condition, participants sat in a standard chair. Cooling in VEST and PASS conditions continued until T_RE_ reached 38.25 °C.

### 2.4. Outcomes

Cooling rate and time to reach T_RE_ of 38.25 °C in VEST and PASS conditions were calculated. Cooling rates were calculated using the following equation:$$Cooling\;rate\;\left(℃\;\min^{-1}\right)=\frac{Initial\;T_{RE}℃-38.25℃}{Time\;\left(\min\right)}$$

### 2.5. Analysis

All statistical analyses were performed using SPSS Statistics version 22 (IBM Corporation, Armonk, NY, USA). An a priori power analysis with an alpha of 0.05, beta of 0.80, and an effect size of 0.8 estimated that 12 subjects were needed in order to achieve a power of 0.83. Normality of cooling rates data by type (VEST, PASS) and sex (male, female) were verified using Shapiro–Wilk test. Data are presented as mean ± SD unless otherwise specified. Paired *t*-tests were used to assess differences between PASS and VEST in exercise duration and post-exercise T_RE_. A two-way analysis of variance was conducted on the influence of cooling modality type (VEST, PASS) and sex (male, female) on cooling rate and cooling time to achieve T_RE_ of 38.25 °C. Partial eta-squared (η^2^_p_) is interpreted as small (0.02), medium (0.13), or large (0.26) [16]. Mean difference (MD) and 95% confidence interval (CI) are provided where applicable. Pearson’s correlation analysis was used to assess the relationships between body mass and body fat percentage on cooling rates for PASS and VEST, separately. The strength of correlation was defined as: (very weak, |0.00–0.19|; weak, |0.20–0.39|; moderate, |0.40–0.59|; strong, |0.60–0.79|; very strong, |0.80–1.0|) [17]. The significance level was set a priori at *p* < 0.05.

## 3. Results

Eight male (mean ± SD; age, 25 ± 4 years; height, 181.1 ± 7.4 cm; body mass, 86.7 ± 10.5 kg; body fat, 16.5 ± 5.2%; body surface area, 2.06 ± 0.15 m^2^) and six female (22 ± 2 years; 163.5 ± 6.7 cm; 61.3 ± 6.7 kg; 22.8 ± 4.4%; body surface area, 1.66 ± 0.11 m^2^) participants completed the study. Post-exercise T_RE_, cooling rate, and cooling time to achieve T_RE_ of 38.25 °C in PASS and VEST by sex is summarized in Table 2. There were no differences in exercise duration (PASS = 48.4 ± 7.1 min, VEST = 51.6 ± 6.8 min, *p* = 0.23) or post-exercise T_RE_ (PASS = 39.63 ± 0.40 °C, VEST = 39.55 ± 0.36 °C, *p* = 0.63) between trials. There was a main effect of cooling modality type on cooling rates (F[1, 24] = 10.46, *p* < 0.01, η^2^_p_ = 0.30), with a greater cooling rate observed in VEST (0.06 ± 0.02 °C·min^−1^) than PASS (0.04 ± 0.01 °C·min^−1^) (MD = 0.02, 95% CI = [0.01, 0.03]) (Figure 2). However, no main effect of sex (F[1, 24] = 0.80, *p* = 0.38, η^2^_p_ = 0.03; MD = 0.01, 95% CI = [−0.01, 0.02]) and interaction effect between cooling modality type and sex (F[1, 24] = 1.71, *p* = 0.20, η^2^_p_ = 0.07) were observed on cooling rates.

There was a main effect of sex (F[1, 24] = 5.97, *p* = 0.02, η^2^_p_ = 0.20) on cooling duration, with a faster cooling time in female (26.9 min) than male participants (42.2 min) (MD = 15.3 min, 95% CI = [2.4, 28.2]). Additionally, a main effect of cooling modality type (F[1, 24] = 4.38, *p* = 0.047, η^2^_p_ = 0.15; MD = 13.1 min, 95% CI = [0.2, 26.0]), and however, no interaction effect between cooling modality type and sex (F[1, 24] = 1.95, *p* = 0.18, η^2^_p_ = 0.08) were observed on cooling duration. Body mass was moderately negatively correlated with the cooling rate in PASS (r = −0.58, *p* = 0.03) but weakly correlated with no statistical significance in VEST (r = −0.25, *p* = 0.39). Body fat percentage showed very weak correlation with no statistical significance in PASS (r = −0.015, *p* = 0.96) and VEST (r = −0.024, *p* = 0.93) cooling rates. Lastly, body surface area was moderately negatively correlated with the cooling rate in PASS (r = −0.53, *p* = 0.05), albeit no statistical significance, and a very weak correlation with no statistical significance was observed in VEST cooling rates (r = −0.18, *p* = 0.53).

## 4. Discussion

The current study investigated the cooling rate of CAERvest^®^ when applied on healthy individuals following exercise-induced hyperthermia (T_RE_, 39.59 ± 0.38 °C) in a hot condition (ambient temperature, 39.8 ± 1.9 °C; relative humidity, 37.4 ± 6.9%). Key strengths of this study include the study design that allowed for uniform environmental conditions across trials and evaluation of CAERvest^®^ cooling rates in both male and female participants. While CAERvest^®^ resulted in a significantly greater cooling rate than passive cooling (VEST, 0.06 ± 0.02 °C·min^−1^; PASS, 0.04 ± 0.01 °C·min^−1^), it did not demonstrate a sufficient cooling rate recommended for EHS treatment (>0.15 °C·min^−1^). Furthermore, participant sex (male, female), body mass (range, 52.9–97.7 kg), body fat percentage (range, 8.9–27.9%), and body surface area (range, 1.49–2.25 m^2^) did not influence the rate of cooling using CAERvest^®^, suggesting that differences in anthropometric characteristics do not provide additional benefit from CAERvest^®^.

Our findings support prior work investigating the efficacy of cooling vests on body cooling in individuals, in that the cooling rate we observed (0.06 °C·min^−1^) is in line with the cooling rates observed previously (range, 0.03–0.05 °C·min^−1^) [9]. Cooling vests provide a practical solution for body cooling in athletics due to their portability and ease of preparation [6]. Other cooling modalities with similar cooling rates include cold (4 °C) intravenous fluid infusion (0.07 ± 0.01 °C·min^−1^) [19] and rotating ice-wet towels on the head, neck, torso, and extremities (0.06 ± 0.02 °C·min^−1^) [20]; the former option can be limited by the presence of medical personnel and the latter option requires a continuous supply of ice-wet towels. Nevertheless, due to limited body surface area that can be cooled at a given time and the reliance on conductive cooling capacity, cooling vests from various manufacturers studied in the previous literature [3,8,9] and current study demonstrated inferior cooling rates compared to whole-body water immersion, and therefore remains a suboptimal cooling method for EHS treatment [2].

Rapid cooling to reduce patient’s internal body temperature below 39 °C within the first 30 min of collapse is a critical aspect of EHS prehospital care [2]. This has been shown to maximize EHS patients’ survival [5] and minimize organ damage from prolonged heat stress [21]. In the case studies of patients who were cooled using cooling modalities with suboptimal cooling rates (<0.15 °C·min^−1^), it was reported that patients experienced sequela (e.g., rhabdomyolysis, ischemic bowel, bilateral compartment syndrome, acute kidney injury, acute liver failure, and disseminated intravascular coagulopathy) requiring hospitalization [21,22,23,24]. Therefore, selection of appropriate cooling methods can dictate the outcome of EHS treatment and clinicians should recognize limitations of cooling methods other than whole-body cold-water immersion (e.g., cooling vests, ice packs, intravenous infusion, and gastric lavage) as a modality of EHS treatment. When compared to a cohort of EHS patients (*n* = 274) with an average initial T_RE_ of 41.4 °C (max T_RE_ of 42.8 °C), patients were successfully cooled in an average of 11.4 min (~18 min with T_RE_ of 42.8 °C) when cold-water immersion was utilized as the cooling modality [5]. If we were to extrapolate the findings from our study to this dataset, the results from our study with the CAERvest^®^ (average cooling rate of 0.06 °C·min^−1^), it would require 41 min to cool a patient from a T_RE_ of 41.4 °C and 64 min from a T_RE_ of 42.8 °C [5], far longer than the 30-min timepoint for optimal survival.

Interestingly, a case of series of patients treated using CAERvest^®^ demonstrated a much higher cooling rate than the current laboratory study (0.13 °C·min^−1^) [8]. This is likely due to the usage of combined cooling modality (e.g., ice packs, cooled intravenous infusion), and the natural convective heat loss given the relatively cool environmental conditions observed at the nearest weather station on the day of collapse (Patient 1 at the 2015 Brighton Marathon, a.m. temperature high in 12 °C; Patient 2 and 3 at the 2015 London Marathon, a.m. temperature high in 16 °C) [25,26]. The findings from the current study suggest that application of the CAERvest^®^ alone, especially in hot environmental conditions, will be insufficient in achieving the cooling rates observed in the prior work [8], thus warranting clinicians to interpret these previous findings carefully.

While CAERvest^®^ alone did not produce an acceptable cooling rate and should not replace whole-body cold-water immersion, the transportable and compact nature of the vest lends itself towards use when transport to a more expedient cooling method is required. For example, when an EHS occurs far from a medical care station, such as several miles away from the marathon finish line, it could be reasonable to use the vest while the patient is being transported to a location where more aggressive cooling (e.g., whole-body cold-water immersion, tarp-assisted cooling) can be provided.

The current study is not without limitations. First, although the normality of data was verified in comparisons of cooling rates by group (VEST, PASS) and sex (male, female), the sample size was relatively small (*n* = 14; male, 8: female, 6). Second, the luteal phase of female participants was determined by self-reported menstrual history and it was not verified by hormonal profile. Third, the influence of anthropometric variables could not be examined when fixed for a certain variable (i.e., examining the impact of body mass when body fat was fixed for sex comparison), due to a wide range of body mass (50.9–96.8 kg), body fat (8.9–27.9%) and body surface area (1.49–2.25 m^2^) observed in our study participants. Lastly, although the manufacturer claims that the vest maintains its cooling effect for up to one hour and absorbs approximately 843 kJ heat energy during this period, we did not measure the surface temperature of cooling vest during our experimental trials.

## 5. Conclusions

In conclusion, body cooling using CAERvest^®^ resulted in a cooling rate that is far below the acceptable standard for EHS treatment following exercise in a hot environment. Therefore, clinicians should not replace whole-body cold-water immersion with CAERvest^®^ when treating EHS in prehospital setting.

## Figures and Tables

**Figure 1 medicina-56-00539-f001:**
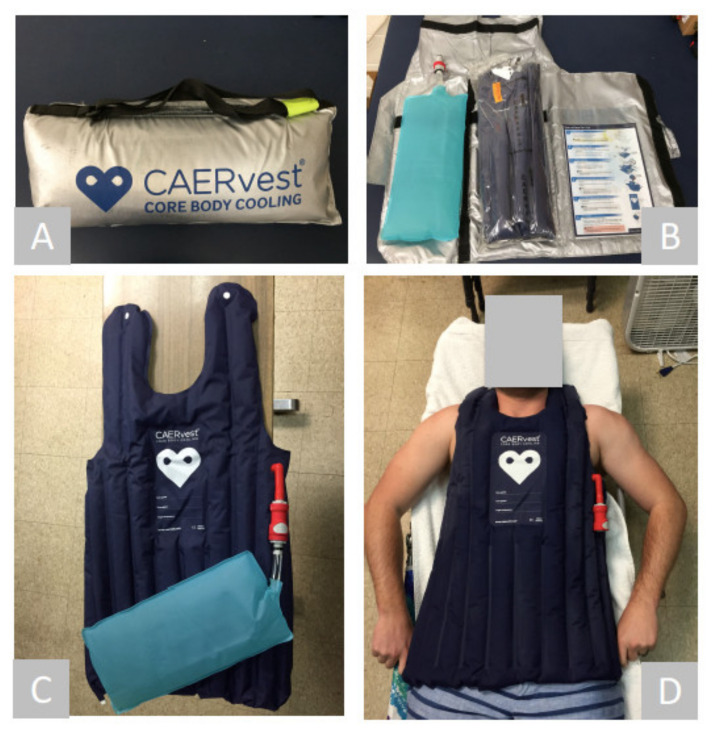
Overview of CAERvest^®^ cooling vest. (**A**) Exterior packaging. (**B**) Contents of packaging. (**C**) CAERvest^®^ being activated using a water reservoir attachment. (**D**) Participant being cooled.

**Figure 2 medicina-56-00539-f002:**
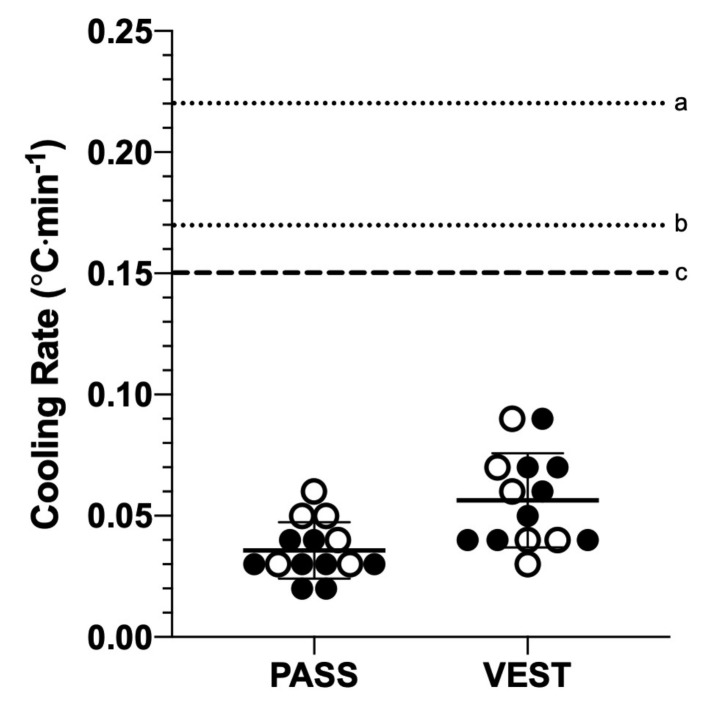
Cooling time to reach rectal temperature 38.25 °C in cooling vest (VEST) and passive cooling (PASS) (*p* = 0.002). Each point represents each individual in corresponding cooling intervention. Horizontal lines represent the mean and standard deviation of cooling time. Closed (●) and open (○) circles represent male and female participants, respectively. Dotted line a, average cooling rate of whole-body cold-water immersion reported by DeMartini et al. [5]; Dotted line b, average cooling rate of tarp-assisted cooling method reported by Hosokawa et al. [18]; Dashed line c, minimal required cooling rate for exertional heat stroke patients.

**Table 1 medicina-56-00539-t001:** Summary of exertional heat stroke patient characteristics reported by Gomm et al. [8].

Patient	Mechanism of Injury	Cooling Modality		Outcome
		Type	Location	
Patient 1	Collapsed at the finish line of the 2015 Brighton Marathon	Ice packs	Groin	Discharged home the same day
		CAERvest^®^	Torso	
		Cooled intravenous solution	Intravenous infusion	
Patent 2	Collapsed at the 2015 London Marathon	Ice packs	Unspecified	Discharged home the same day
		CAERvest^®^	Torso	
Patient 3	Collapsed at the finish line of the 2015 London Marathon	CAERvest^®^	Torso	Rhabdomyolysis, intestinal ischemia, and bilateral compartment syndrome

**Table 2 medicina-56-00539-t002:** Summary of post-exercise T_RE_, cooling rate, and cooling time to achieve T_RE_ of 38.25 °C in PASS and VEST by sex (mean ± standard deviation).

	Cooling Modality	Sex	
		Male	Female
Post-exercise T_RE_ (°C)	PASS	39.73 ± 0.41	39.49 ± 0.39
	VEST	39.76 ± 0.27	39.29 ± 0.30
Cooling rate (°C·min^−1^)	PASS	0.03 ± 0.01	0.04 ± 0.01
	VEST	0.06 ± 0.02	0.06 ± 0.02
Cooling time (min)	PASS	53.1 ± 23.4	29.0 ± 9.5
	VEST	31.2 ± 12.7	24.7 ± 14.4

Abbreviations: T_RE_, rectal temperature; PASS, passive cooling trial; VEST, cooling vest trial.

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
