# Peer review of "Chemically Activated Cooling Vest’s Effect on Cooling Rate Following Exercise-Induced Hyperthermia: A Randomized Counter-Balanced Crossover Study"

_medicina, 2020, doi:10.3390/medicina56100539_

Round 1

Reviewer 1 Report

This is a very interesting study, well designed and conducted.  It has being well wrote, with good scientific support. There is only one observation, there is no biological plausibility about an association between gender and heat strain. The amount of heat stored in the body is related with body mass, and fat, therefore as women in general have a smaller body mass than men, we can expect that they have faster cooling time.

Taking this consideration into account, I suggest not to mention that the results are related with gender.

Author Response

Reviewer 1

This is a very interesting study, well designed and conducted.  It has being well wrote, with good scientific support.

Response: We thank the reviewer for their positive comments and review.

There is only one observation, there is no biological plausibility about an association between gender and heat strain. The amount of heat stored in the body is related with body mass, and fat, therefore as women in general have a smaller body mass than men, we can expect that they have faster cooling time. Taking this consideration into account, I suggest not to mention that the results are related with gender.

Response: Thank you for your comments. We acknowledge that sex in itself is not the reason for faster cooling rate, but it is due to the inherent difference in their anthropometric characteristics. We have modified our secondary aim sentence in Line 97 to clarify this point.

Reviewer 2 Report

 This interesting study aimed to examine if a portable cooling vest meets the established cooling criteria for exertional heat stroke treatment. The article is well written; Unfortunately, there are some inaccuracies that need to be corrected.

Line 21: there is a big difference between the weights of males and females. was this difference selected a priori? please, comment.

Line 24: It seems to me an ethically sensitive trial. Please, make sure that the reader can access (via link) the acceptance of the ethics committee.

Line 29: This result is obvious given the weight difference between men and women. I propose to consider a further variable, (cooling duration / mass).

Line 61: This table can be deleted, reporting the relative data in text.

Line 94: girls appear with more fatty tissue than boys. This can skew the data on the male-female difference. Please, comment.

Line 94: I think the inclusion criteria have not been sufficiently clarified. For example, I think the amount of fat in athletes who actually undergo heatstroke is much lower than the sample considered in this study. Clarify this criticality.

Line 103: Insert the link where the reader can see the acceptance of the ethics committee.

Line 150: This formula assumes a linear trend. Theoretically, perhaps it is better to think of a negative exponential. Please, comment.

Line 161: “The strength of correlation was defined as: (very weak, |0.00-0.19|; weak, |0.20-0.39|; moderate, |0.40-0.59|; strong, |0.60-0.79|; very strong, 161 |0.80-1.0|)”. Please, add reference.

Line 162: Power analysis is completely missing. It is essential to insert it.

Line 165: It is unlikely to measure down to hundredths of a degree. Please, report the exact number of significant digits.

Line 164-182: Report, for each test applied, the appropriate effect size value. Also include the explanation in material and methods.

In general, I ask that the authors follow the "consort" protocol.

Author Response

Reviewer 2

This interesting study aimed to examine if a portable cooling vest meets the established cooling criteria for exertional heat stroke treatment. The article is well written; Unfortunately, there are some inaccuracies that need to be corrected.

Response: We thank the reviewer for their positive comments and suggestions to improve the paper.

Line 21: there is a big difference between the weights of males and females. was this difference selected a priori? please, comment.

Response: We did not recruit participants using a priori selection criteria for body stature.

Line 24: It seems to me an ethically sensitive trial. Please, make sure that the reader can access (via link) the acceptance of the ethics committee.

Response: Current study was approved by the Institutional Review Board at the University of Connecticut (Line 117). Unfortunately, the University of Connecticut IRB does not have an IRB repository for public access. It should be noted that due to the nature of the study, all trials were conducted under the supervision of trained healthcare professional.

Line 29: This result is obvious given the weight difference between men and women. I propose to consider a further variable, (cooling duration / mass).

Response: Thank you for your comment.  Although we agree that the difference in cooling duration between sex may have been obvious due to their body size difference in our cohort, we believe that it is one of the key findings of this study and would like to keep this information in the abstract.

We also calculated cooling duration/mass for both sexes in VEST, as suggested by the reviewer. Values for male and female were 0.36 and 0.39, respectively, and no difference was observed statistically using t-test (p=0.729).

Line 61: This table can be deleted, reporting the relative data in text.

Response: Thank you for your comment. Authors believe that the information summarized in Table 1 is better conveyed to readers using the current format and would like to keep this section as is.

Line 94: girls appear with more fatty tissue than boys. This can skew the data on the male-female difference. Please, comment.

Response: Thank you for your comment. We have added the following statement as a limitation of this study.

“Third, the influence of anthropometric variables could not be examined when fixed for a certain variable (i.e., examining the impact of body mass when body fat was fixed for sex comparison) due to a wide range of body mass (50.9–96.8 kg), body fat (8.9–27.9%) and body surface area (1.49–2.25m2)observed in our study participants.”

Line 94: I think the inclusion criteria have not been sufficiently clarified. For example, I think the amount of fat in athletes who actually undergo heatstroke is much lower than the sample considered in this study. Clarify this criticality.

Response: Thank you for your comment. Studies from race events (1,2) attest that exertional heat stroke can be observed in non-elite, recreationally active individuals. Our recruitment criteria (i.e., individuals who engage in at least 30 minutes of exercise 4–5 days per week) was used with a foresight to individuals who are close to those participating in race events.

  1. Demartini JK, Casa DJ, Stearns R, Belval L, Crago A, Davis R, et al. Effectiveness of cold water immersion in the treatment of exertional heat stroke at the Falmouth Road Race. Med Sci Sports Exerc 2015;47:240–5. https://doi.org/10.1249/MSS.0000000000000409.
  2. Hawes R, McMorran J, Vallis C. Exertional heat illness in half marathon runners: experiences of the Great North Run. Emergency Medicine Journal. 2010;27(11):866-867. doi:10.1136/emj.2010.090928

Line 103: Insert the link where the reader can see the acceptance of the ethics committee.

Response: Unfortunately, the University of Connecticut IRB does not have an IRB repository for public access.

Line 150: This formula assumes a linear trend. Theoretically, perhaps it is better to think of a negative exponential. Please, comment.

Response: Thank you for your comment. For the purpose of our current paper, we have used the formula that assumes a linear trend so that the cooling rates can be compared from previous studies.

Line 161: “The strength of correlation was defined as: (very weak, |0.00-0.19|; weak, |0.20-0.39|; moderate, |0.40-0.59|; strong, |0.60-0.79|; very strong, 161 |0.80-1.0|)”. Please, add reference.

Response: Citation (Evans, 1996) was added to the main text and reference list.

Line 162: Power analysis is completely missing. It is essential to insert it.

Response: Thank you for your comment. Following statement was added to the analysis section of the method (Lines 168–171). “An a priori power analysis was conducted using sealed envelope™ (Sealed Envelope Ltd. 2012) to test the difference between two independent group means using a one-tailed test with 0.05 alpha level and desired power level of 0.80. Result showed that a total sample of size of six was required.”

Line 165: It is unlikely to measure down to hundredths of a degree. Please, report the exact number of significant digits.

We have modified all duration (minutes) related data to report down to the 0.1 minute.

Line 164-182: Report, for each test applied, the appropriate effect size value. Also include the explanation in material and methods.

Partial eta-squared is now added to the two-way ANOVA findings and an explanation on thresholds used for small/medium/large effect is added to the method section.

Note: Although the overall conclusion was not affected, we have realized that some of the statistics reported in the original submission was incorrect. All data were verified

In general, I ask that the authors follow the "consort" protocol.

Thank you for your comment. By addressing all of the above comments thus far, we believe we have addressed items outlined in CONSORT.

Round 2

Reviewer 2 Report

I have read the corrected version of the authors; although some small improvements have been made, the authors did not fully respond to serious limitations in the paper.

See below:

Old question: “Line 103: Insert the link where the reader can see the acceptance of the ethics committee.

Response: Unfortunately, the University of Connecticut IRB does not have an IRB repository for public access.”

Comment: Ethic committee is mandatory and the documents of the ethics committee must be accessible!

Old question: “Line 162: Power analysis is completely missing. It is essential to insert it.

Response: Thank you for your comment. Following statement was added to the analysis section of the method (Lines 168–171). “An a priori power analysis was conducted using sealed envelope™ (Sealed Envelope Ltd. 2012) to test the difference between two independent group means using a one-tailed test with 0.05 alpha level and desired power level of 0.80. Result showed that a total sample of size of six was required.””

Comment: The calculation shown here is wrong and incorrectly written. What is the difference between the expected averages defined a priori in order to obtain the reported sample size? Why is a one-tailed test used? Why if the calculated sample size is = 6, then a higher number of patients is enrolled?

Old question: “In general, I ask that the authors follow the "CONSORT" protocol.”

Response:Thank you for your comment. By addressing all of the above comments thus far, we believe we have addressed items outlined in CONSORT.

Comment: Indeed, many points of the CONSORT protocol have not been satisfied.

Author Response

I have read the corrected version of the authors; although some small improvements have been made, the authors did not fully respond to serious limitations in the paper.

See below:

Response #2: We thank Reviewer 2 for additional comments, which we have addressed in the revised manuscript as discussed below in bold.

Old question: “Line 103: Insert the link where the reader can see the acceptance of the ethics committee.

Response: Unfortunately, the University of Connecticut IRB does not have an IRB repository for public access.”

Comment: Ethic committee is mandatory and the documents of the ethics committee must be accessible!

Response #2:

 We appreciate the comment from the reviewer. The ethics committee, in this case the Institutional Review Board at the University of Connecticut, approved the study. We have modified the text to clarify that the ethics committee approving this study is called the Institutional Review Board (https://ovpr.uconn.edu/services/rics/irb/) at the institution where the study was conducted (Lines 117-119 of the revised text). We have listed the protocol ID number (H15-153) for the readers to reference and we would like to reiterate that the granting ethics committee at the institution where the study was conducted does not provide a publicly accessible repository for viewing of approved protocols. Since there is no publicly available repository for the reader to view the approval, it is possible for the authors to submit a copy of the ethics committee approval letter as a supplementary file, however, we will leave this decision up to the academic editor to decide. In addition, we would like to point out that according to the author guidelines for Medicina, authors are not required to submit a publicly accessible link for ethics approval within the manuscript and mentioning of ethics approval and protocol number should be sufficient.

Old question: “Line 162: Power analysis is completely missing. It is essential to insert it.

Response: Thank you for your comment. Following statement was added to the analysis section of the method (Lines 168–171). “An a priori power analysis was conducted using sealed envelope™ (Sealed Envelope Ltd. 2012) to test the difference between two independent group means using a one-tailed test with 0.05 alpha level and desired power level of 0.80. Result showed that a total sample of size of six was required.””

Comment: The calculation shown here is wrong and incorrectly written. What is the difference between the expected averages defined a priori in order to obtain the reported sample size? Why is a one-tailed test used? Why if the calculated sample size is = 6, then a higher number of patients is enrolled?

Response #2:

Thank you for the comment. We have amended the discussion of the power analysis at the request of the reviewer. Also, for clarification purposes, we used a one-tailed test because we hypothesized that the cooling vest would have a cooling rate higher than that of passive rest. 

Old question: “In general, I ask that the authors follow the "CONSORT" protocol.”

Response: Thank you for your comment. By addressing all of the above comments thus far, we believe we have addressed items outlined in CONSORT.

Comment: Indeed, many points of the CONSORT protocol have not been satisfied.

Response #2:

Please see our response to each of the items include in the CONSORT checklist.

Title and abstract

  • Identification as a randomised trial in the title
    • Title was modified to include “A Randomized Counter-balanced Crossover Study” (Lines 4-5)
  • Structured summary of trial design, methods, results, and conclusions
    • Description of study design was added to the abstract (Lines 28-29)

Introduction

  • Scientific background and explanation of rationale
    • Scientific background and explanation of study rationale can be found in Lines 43-76 and Lines 82-96, respectively.
  • Specific objectives or hypotheses
    • Specific objectives and hypotheses can be found in Lines 97-103.

Methods

  • Description of trial design (such as parallel, factorial) including allocation ratio
    • Study design is explained in Lines 106-107.
  • Important changes to methods after trial commencement (such as eligibility criteria), with reasons
    • None
  • Eligibility criteria for participants
    • Eligibility criteria of participants are explained in Line 111(i.e., recreationally active) and Lines 114-117.
  • Settings and locations where the data were collected
    • Settings and locations of data collection are explained in Lines 107-109.
  • The interventions for each group with sufficient details to allow replication, including how and when they were actually administered
    • A thorough explanation of procedures completed by participants and description of trials can be found in Lines 122-173.
  • Completely defined pre-specified primary and secondary outcome measures, including how and when they were assessed
    • Primary study outcome (cooling rate) is defined in Lines 179-1181.
  • Any changes to trial outcomes after the trial commenced, with reasons
    • None
  • How sample size was determined
    • Sample size justification can be found in Lines 184-185.
  • When applicable, explanation of any interim analyses and stopping guidelines
    • Not applicable
  • Method used to generate the random allocation sequence
    • Method used to generate the random allocation sequence was added in Line 127.
  • Type of randomisation; details of any restriction (such as blocking and block size)
    • Study design is explained in Lines 106-107.
  • Mechanism used to implement the random allocation sequence (such as sequentially numbered containers), describing any steps taken to conceal the sequence until interventions were assigned
    • Method used to generate the random allocation sequence was added in Line 127.
    • Concealment (blinding) of the trial was not possible in current study due to the nature of intervention (i.e., application of cooling vest).
  • Who generated the random allocation sequence, who enrolled participants, and who assigned participants to interventions
    • First author of the paper (YH) generated the random allocation sequence, enrolled participants, and assigned participants to interventions. This information has been added to Line 127.
  • If done, who was blinded after assignment to interventions (for example, participants, care providers, those assessing outcomes) and how
    • Not applicable
  • If relevant, description of the similarity of interventions
    • Not relevant
  • Statistical methods used to compare groups for primary and secondary outcomes
    • Statistical methods used in the study are explained in Lines 183-196.
  • Methods for additional analyses, such as subgroup analyses and adjusted analyses
    • Not applicable

Results

  • For each group, the numbers of participants who were randomly assigned, received intended treatment, and were analysed for the primary outcome
    • The numbers of participants included in the study were moved from method section to result section (Lines 198-200).
  • For each group, losses and exclusions after randomisation, together with reasons
    • None
  • Dates defining the periods of recruitment and follow-up
    • The period of recruitment and data collection was added in Lines 119-120.
  • Why the trial ended or was stopped
    • At the completion of all intended trials, the study was closed.
  • A table showing baseline demographic and clinical characteristics for each group
    • It is reported in sentence, in Lines 198-200.
  • For each group, number of participants (denominator) included in each analysis and whether the analysis was by original assigned groups
  • For each primary and secondary outcome, results for each group, and the estimated effect size and its precision (such as 95% confidence interval)
    • All results are reported in suggested format (Lines 202-219).
  • For binary outcomes, presentation of both absolute and relative effect sizes is recommended
    • Not applicable
  • Results of any other analyses performed, including subgroup analyses and adjusted analyses, distinguishing pre-specified from exploratory
    • Not applicable
  • All important harms or unintended effects in each group (for specific guidance see CONSORT for harms)
    • Not applicable

Discussion

  • Trial limitations, addressing sources of potential bias, imprecision, and, if relevant, multiplicity of analyses
    • Trial limitations are explained in lines 356-365.
  • Generalisability (external validity, applicability) of the trial findings
    • Generalizability of the trial findings are summarized in Lines 350-355, and Lines 367-369.
  • Interpretation consistent with results, balancing benefits and harms, and considering other relevant evidence
    • Interpretation of results are summarized in Lines 295-349.

Other information

  • Registration number and name of trial registry
    • The study registration number (under University of Connecticut Institutional Review Board) is specified in Lines 118-119.
  • Where the full trial protocol can be accessed, if available
    • It is not publicly accessible.
  • Sources of funding and other support (such as supply of drugs), role of funders
    • Funding source is specified in Line 376.

Round 3

Reviewer 2 Report

the authors responded to most of the limitations highlighted in previous reviews. Even if some aspects could be clarified, also taking into account the positive opinion of the other reviews, I believe that the paper can be published on medicine.